# Integrated Reporting and Integrated Thinking: Proposing a Reporting Model That Induces More Responsible Use of Corporate Power

Guler Aras [1] and Paul F. Williams [2,*]

1. Business Administration Department, Faculty of Economics and Administrative Sciences, Yildiz Technical University, Barbaros Bulvan, 34349 Istanbul, Turkey; aras@yildiz.edu.tr
2. Department of Accounting, Poole College of Management, North Carolina State University, Raleigh, NC 27695, USA
* Correspondence: pfwms@ncsu.edu

**Abstract:** The obligations of corporations to members of society have been problematic since the corporate form came into existence. Under different rubrics, reporting firms' socially responsible behavior has been extensively debated, and researched, for at least the past half century. The latest incarnation of corporate social reporting is labeled integrated reporting—the blending of the traditional financial report with a report on the firms' achievements as socially responsible beings. In this paper, we provide a brief history of corporate social reporting to provide sufficient context for our discussion of a model of integrative reporting that provides for a better representation of just how socially responsible firms are. Progress so far in achieving meaningful integrated reporting that produces more socially responsible corporate citizens is disappointing. The structured narrative of financial performance still dominates the unstructured narrative about social performance. We argue this is partially attributable to two intellectual constraints limiting our ability to imagine systems that could produce better social outcomes from corporate behavior. One constraint is the dominance of "decision usefulness" as the purpose of accounting. The second intellectual constraint is the reluctance to seriously consider that the problem of corporate social responsibility (CSR) lies in the corporate form itself. Thus far, the integration of these reports to give equal status to financial and social performance is not close to achievement. We propose that a first step to developing an integrated report is to adopt a governmental reporting model for corporations. If the six capitals model proposed by IIRC is to be a movement toward more ethical corporate behavior, then the six capitals must be deemed as equally valuable ends and certainly not subservient to only financial ends. The current financial reporting model strongly mitigates against this happening. We argue that each of the capitals is analogous to what in governmental parlance is a "program" or "function", which require the commitment of financial resources for accomplishment. Thus, a truly integrated report will disclose to all stakeholders what resources are committed to enhancing each of the six capitals as ends in themselves.

**Keywords:** integrated reporting; sustainability; content analysis; long-term value creation; six capitals; agency theory; voluntary disclosure theory

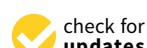



## 1. Introduction

In the last a few decades, many companies have come to realize that meeting stakeholder expectations is necessary for creating long-term sustainable value and achieving strategic business objectives. Creating long-term sustainable value is more important than the short-term maximization of shareholder value, which has become the mantra for top managements in their approach to corporate missions in the last couple of decades. Regulators and accounting standard-setters have focused primarily on the contents of financial statements and the utility of those financial statements to capital providers. Although stakeholder management has become an increasingly important tool for management,

accounting policy and practice, focused as they are on financial reporting practices, are inadequate for meeting stakeholder concerns.

There may have been a time in the past when it was reasonable to assume that the knowledge about corporate conduct conveyed merely by financial information was a sufficient corporate report. However, the debate about the proper role of the corporation in society has intensified. There is the awareness that corporations have been instrumental not just as engines of prosperity, but also as engines of growing global income and wealth inequality [1,2] and major contributors to environmental degradation and the corruption of democratic institutions [3,4]. The most recent stage in the development of corporate reporting epitomizes the recognition of the ramifications of corporate behavior for other than just the financial well-being of stockholders. A growing number of global corporations are no longer merely seeking to communicate to a purely economic constituency but rather their focus is upon a more expansive audience who have a stake in the story a business's activities have to tell. Through what has come to be called sustainability or corporate social responsibility reports, firms are disclosing not only financial information but also non-financial information pertaining to their governance, their social effects, their environmental effects, and company strategies about how to address stakeholder concerns. This integrated approach [5] aims to be more comprehensively informative about firms' actions and, thus, be more responsible to stakeholders other than just shareholders and creditors. The integrated reporting model proposed by IIRC is based on integrated thinking and a comprehensive value creation story of the firm. There are also some motivations to change to this reporting approach in order to garner regulatory support and to alter stakeholder and investor expectations.

Moreover, The United Nation's Sustainable Development 17 Goals have made it clear that the landscape is changing in corporate reporting. The 17 goals reflect a growing awareness that measures singularly focused on economic values are insufficient to explain the value a corporation adds to or takes from the welfare of society. As economists have long noted, GNP is not a welfare measure [6]. Indeed, GNP is numerically enhanced by actions that demonstrably reduce social welfare, e.g., pollution. The traditional method of financial reporting is analogous to GNP, but at the level of the firm. It is not the case that a firm's net income is a measure of its contribution to social welfare. Therefore, the importance of the corporate information contained in its reports has changed. Corporate reporting is viewed as having to address the financial and non-financial information needs of those who must make decisions about a business now that they are situated within this changed business dynamic. The old corporate reporting is based on the assumption that shareholders are owners of corporations and it is they whose interests corporations should exclusively serve. However, given the dual problems of growing inequality and environmental decline, people are being convinced that we need to move away from the shareholder primacy model and re-conceptualize the corporation as a social institution for the advancement of the long-term interests of its stakeholders and all of society.

Much of the discussion about integrated reporting still revolves around a linear model of the economic process. The firm takes energy and materials from the natural environment; makes something with those resources; what is made is in turn used; and finally we lose in that process via wasted heat and wasted matter [7] (p. 180). Nicholas Geogescu-Roegan [8] famously described the economic process as one characterized by entropy, the end result of which is garbage. Georgescu-Roegan's characterization has spawned a field of economics, ecological economics, which rejects many of the basic premises of neoclassical economics upon which are based the proposed integrated reporting models. The integrative reporting models proposed so far still reflect this degenerative model, with financial reports focused solely on the firm's profitability and another set of reports designed to indicate the extent to which the firm is reducing the negative externalities that inevitably result from the economic model that has waste as its ultimate product.

As Raworth [7] (p. 181) characterizes it, nothing short of a new economic model will be required if businesses are to be truly sustainable:

> *"This ubiquitous industrial model has delivered strong profits to many businesses and has financially enriched many nations in the process. But its design is fundamentally flawed because it runs counter to the living world, which thrives by continually recycling life's building blocks such as carbon, oxygen, nitrogen and phosphorous. Industrial activity has broken these natural cycles apart . . . ."*

The idea of integrated reporting anticipates the systems approach to economic design, a regenerative approach to business, but thus far, the tangible results of such reporting fall short of providing the transformative power that is necessary for firms to contribute to a genuinely sustainable economy [9,10].

There are now reports described as integrated but in a superficial way. What we now call integrated reporting is the combining of the traditional financial statements and information about the effects of activities aimed at outcomes other than financial ones. The basis for the integrated reporting model is the idea of the balanced scorecard, which in turn is predicated on the assumption that for firms to survive in the long run, attention must be paid to the environmental and social consequences of firm actions. For example, becoming more energy efficient (reducing carbon footprint) involves significant investment and disruption that, in the short run, can reduce profitability. However, to neglect moving to energy efficiency has the likely consequence of substantially diminishing a firm's long-term vitality. Information in integrated reports that describes the results of energy efficiency efforts is fine as far as it goes, but as yet firms have not actually "integrated" such information. Putting an amount of copper and an amount of zinc into the same container yields merely a container filled with copper and zinc. Integrating those two elements changes their respective properties into that of their alloy brass. If corporations are to be institutions contributing to the solutions to the major problems besetting society, then their story has to be constructed not just of amounts of copper and amounts of zinc, but of brass—this seems to be what is required to tell a coherent story about corporate activity. The purpose of our analysis in this paper is to outline an integrated reporting model that takes into account the severe weaknesses of a reporting model based on "decision usefulness" in order to tell a truer business story for stakeholders and all of society. Ours is a model based on the idea that corporate reports are reports aimed at consummating accountability relationships [11,12]. A truly integrated reporting system conveys a story about how a company contributes to social well-being and not how it enhances shareholder value. An accountability model acknowledges that the very idea of corporate social responsibility presumes corporations should behave morally and that moral reasons are legitimate reasons for corporate decisions [13–15] and are not solely responsible for creating private wealth but for public wealth as well. This implies that good management requires assuming a more extensive sense of moral agency than required by the maximization of shareholder value. We show how the accountability model applied to governments can be adapted to corporations in order to reveal what value is devoted to achieving sustainability objectives.

The following section provides a brief history of corporate social reporting to provide sufficient context for our discussion of an accountability model of integrative reporting that provides for a better representation of just how socially responsible firms are. The third section proposes a reporting model underlining the significance of integrative thinking. The fourth section presents critical discussions, while the final section pinpoints the conclusions for theory, the implications for public institutions, and the limitations and further research directions.

## 2. From Social Audit to Integrated Reporting

Since the mid-nineteenth century and the start of the Industrial Revolution there has been ongoing disputation about the proper role of the industrial corporation in society. With the publication of Berle and Meanes' [16] classic work on the modern corporation and its discussion of the separation of management from ownership, there is the yet-unresolved question of what the proper role of management is; what that role is shifts with the political wind. One role that has been in ascendance for the past four decades is the "classical" role

for management as agents of the corporation's owners—its shareholders. The classical expression of this role, enunciated famously by Milton Friedman [17], is to make as much profit for the shareholders as is allowed within the rules of the game. Since this enunciation, the idea of profit maximization has shifted to become that of shareholder value maximization, which is not necessarily the same thing, as far as management behavior is concerned. Theoretically, a share price reflects investors' expectations about future profitability, but in reality, share prices are affected in the short run by other considerations, e.g., irrational exuberance [18]. Behaviors on the part of management that focus on shareholder value (share price) are by nature short-term, since managers are punished in the short run for failure to meet market expectations about short-term earnings. So, even under the classic notion of managerial responsibility, there is ambiguity about what those responsibilities are and what the appropriate managerial behavior is.

Danley [19] labels an alternative narrative about the role of the corporation and its managers "managerialism." Managerialism is predicated on the idea that management is a profession, which equips its practitioners with tools of scientific management. Managers are thus economic statesmen who manage the corporation, fully cognizant of the effects the corporation has on multiple constituents (stakeholders), and balance the legitimate interests of those various constituents to achieve some socially acceptable optimum. In the U.S., this image of management gained considerable momentum during the 1960s and early 1970s.

A typical expression of what constituted managerialism was expressed by the Committee on Economic Development [20] (p. 22):

> *"The modern professional manager also regards himself, not as an owner disposing of personal property as he sees fit, but as a trustee balancing the interests of many diverse participants and constituents in the enterprise, whose interests sometimes conflict with those of others".*

The rationale provided for this view of management's role is "enlightened self-interest":

> *"There is a broad recognition today that corporate self-interest is inexorably involved in the well-being of the society of which business is an integral part, and from which it draws the basic requirements needed for it to function at all—capital, labor, customers (sic). There is increasing understanding that the corporation is dependent on the goodwill of society, which can sustain or impair its existence through public pressures on government. And it has become clear that the additional resources and goodwill of society are not naturally forthcoming to corporations whenever needed, but must be worked for and developed."* [20] (p. 27).

During the 1970s, this managerialism narrative about the role of corporate management was labeled "corporate social responsibility" [21–24]. A corollary development to managerialism was advocacy for a form of expanded reporting on management performance via the creation of a system of reporting on corporate efforts to fulfill social responsibilities. This is commonly referred to as corporate social reporting (CSR) or the social audit. Such reports were incorporated into the usual financial reports or were issued as separate reports, which contained information regarding the corporation's efforts to contribute to allaying whatever were the compelling social issues of the day. In the 1970s, the environment was a prominent issue because scientists were pointing to the deterioration of the physical environment brought about by industrial production, most notably via the report of the Club of Rome [25]. Other prominent issues of the day were business with South Africa and its apartheid system and involvement with the weapons industry, etc. What was considered a pertinent corporate focus for affecting responsible conduct changed as new issues emerged and old issues ceased being relevant. For example, the apartheid system in South Africa ended, but animal rights emerged as a new issue that affected companies that relied on animal testing for developing their products. To expect corporations to be socially responsible implies that corporations are moral agents [13] and

that managers' reasons for corporate actions include moral reasons, not just those based on financial calculations of profit and loss.

Somewhat ironically, the impetus for managerialism and CSR was dissipated by the actions of business to undo the New Deal consensus that prevailed in the U.S. through the 1960s. Under the leadership of Hayek's Mont Pelerin Society and the University of Chicago, a political movement named by its founders as neoliberalism [26–28] gained momentum with the assistance of the corporate community, most notably by the Business Roundtable. The stagflation of the late 1970s provided the opportunity for neoliberal economic policies to gain a foothold, and the elections of Ronald Reagan as U.S. president and Margaret Thatcher as P.M. in the U.K. established neoliberal ideology as the governing consensus in those countries until the present. Neoliberal ideology has spread around the world as even the putatively communist nation of mainland China has adopted neoliberal economic policies and market logic.

The hegemony of neoliberal ideology in most advanced economies has restored the classical view of management. Supported by neoliberal economic theories of efficient capital markets and principal/agent theory [29], the predominant narrative about management is currently that managers are agents of the shareholders and have as their primary responsibility the maximization of shareholder value [30,31]. Neoliberalism is built on a faith in markets and the wisdom of prices as the guide to the conduct of human affairs. Concomitant with a faith in markets is an aversion to regulation, which allegedly inhibits the free working of markets. However, the problems that led to managerialism and CSR have persisted; neoliberal ideology has failed to address these problems effectively and has arguably made the problems worse [32]. As the follow up study to the Club of Rome report [33] indicated, the environmental problems identified in the 1960s have gotten worse and many scientists are predicting a grim future if immediate action is not taken to curb man-made climate change. Income and wealth inequality have grown substantially [1] in most Western democracies. The failure of faith in the self-regulating property of markets has revivified interest in a managerial narrative that anticipates a broader remit for management behavior than entertained by the classical role confined to maximizing shareholder value.

Thus, the corporate social responsibility of the 1970s has emerged with renewed energy but is now more frequently referred to as corporate sustainability. It is an acknowledgment that for corporations to be viable, their behavior must be such as to leave for subsequent generations the same choices and opportunities as those that are available to the current generation [34]. Sustainability is not simply confined to issues of recurring access to necessary natural resources but has been described as a three-legged stool consisting of corporate actions that are not only environmentally responsible but economically and socially responsible as well [35]. Leading management scholars have described the situation thusly:

> " . . . *never before have we seen the speed, extent, and magnitude of resource loss that we observe now. Whether it is soil, water, nutrition, a stable climate, or social equity as measured by the rich-poor gap, the list of declining resources in question is relevant for nearly the entire global economy, with no company left unaffected. And that, in turn, creates a fundamental change in how companies compete to create enduring value.*" [36] (pp. 9–10).

Echoing the CED narrative from 1971, these scholars see the remedy in a new paradigm for business and corporate action:

> "*We are committed to sharing an exciting but largely invisible story of a shift in the conduct of business. In the **new narrative** (emphasis in original), the gloom and doom of declining resources is also the foundation for opportunity, an emerging paradigm of business that can be **more sustainable** (emphasis in original) and profitable.*" [36] (p. 9).

Under the rubric "sustainability" has come intense interest in methods of reporting on management efforts and accomplishments for making firms more sustainable. Numerous

organizations have been formed to develop systems of sustainability reporting, e.g., GRI, and the Sustainability Accounting Standards Board. Currently, the most recognized is the Global Reporting Initiative (GRI), which has provided guidelines on what types of activities sustainable companies should report. The framework proposed by GRI emphasizes reporting along specific dimensions generally associated with responsible corporate behavior. The general terms describing these dimensions are:

- Environment—activities that pertain to a company's efforts to reduce its adverse effects on the environment;
- Society—activities that benefit the communities in which a company does business, including charitable activities;
- Economic—activities that enhance the economic contributions of the company, such as job creation and value added;
- Labor—activities that contribute to the wellbeing of employees, such as training and benefits;
- Product responsibility—activities that contribute to providing products or services that are safe to use, reliable, and whose use and disposal minimizes the adverse impacts on others;
- Human rights—activities conducted that enhance human rights in all of the countries in which a company may do business;
- Corporate governance—activities that make corporate governance more transparent and effective.

GRI guidelines are not definitive but do indicate the nature of a role for management that is more expansive than the classical role of singular focus on profitability or firm economic value.

### 2.1. Promise as Yet Unfulfilled

Though there has been extensive effort on the part of most large corporations to provide reports about their sustainability activities and considerable scholarly literature on these efforts, there is still skepticism about sustainability reporting's purpose and success [10,37,38]. As Milne [39] (p. 143) speculates on the continuing skepticism about sustainability reporting:

> "At the heart of assessing corporate 'sustainability' reporting are fundamental differences about what corporate reporting for sustainability **means** (emphasis in original) and, implicitly within these differences, what purposes it serves (or might serve), and whose interests are (or might be) served by it. What is to be sustained?"

Sustainability is an ambiguous term. In an ecological sense, sustainability means the ability of any species to continue reproducing itself into an indefinite future. Some species have been enormously successful in sustaining their continued presence on Earth, e.g., alligators and crocodiles, related to dinosaurs, have managed to be present on Earth 65 million years longer than their cousins. In this sense of the term, sustainability refers to a way of life that can ensure the continued reproductive success of a species. This sense of sustainability implies a systemic view of ecology in which species have to adapt to changes in a dynamic system governed by certain natural laws that dictate what is possible for any species. This idea of sustainability is captured by Rudyard Kipling's poem about the law of the jungle, whose last stanza reads: "Now these are the Laws of the Jungle, and many and mighty are they;/But the head and the hoof of the Law and the haunch and the hump is—Obey!".

Sustainability efforts by many large, international corporates (e.g., Walmart and Coca Cola) imply a different understanding of sustainability. Sustainability efforts on the part of these companies emphasize what Dauvergne and Lister [40] describe as "eco-business." The sustainability narrative of eco-business is one of being profitable by doing good; it is substantively about obtaining control over vital natural resources. Eco-business focuses on cost reduction and is not a significant departure in thinking from rationales for corporate

social responsibility that were prevalent decades ago. The primary value pursued through eco-business is the customary economic value.

Research aimed at understanding the effects of sustainability/triple bottom line reporting has yet to establish a strong link between what is reported and the actual performance of companies with respect to sustainability activities. The evidence points to such reporting being mainly an effort at public relations and legitimacy. That is, firms focus on appearing to be responsive to the interests of various stakeholders, rather than actually internalizing the various interests of stakeholders as goals [38,41–48]. That such research reaches this conclusion is perhaps not too surprising since the predominant role served by dimensions of sustainable performance such as those of the GRI is mainly as hygiene factors in terms of management strategy and decision-making. A hygiene factor is like a nonbinding constraint. It is a factor to consider, so long as it is feasible to do so. It is not a factor trumping the overarching goal of the company's financial success. Financial success is a very structured narrative about a company. Financial reporting is governed by rules, which allegedly generate "measures" of financial success. Profit is "measured" and constitutes a metric of success that is widely accepted. Financial statements formally prepared at precise intervals track the financial position, earnings, and cash flows of a company in a seemingly objective manner. Historically, the financial reports, for the most part the focus of financial market participants, were reported separately from other kinds of information such as sustainability information because of the rigid structure of such reports. Standard-setting bodies and professional bodies have developed extensive rules about how these reports are to be prepared and what kinds of behaviors those reports are to reflect. Sustainability reports have been developed more-or-less independently of the formality of the financial reports.

*2.2. Value Judgments and the Financial Reporting Model*

This difference in structure—the formality of financial reports versus the informality of sustainability reports—mitigates in favor of management by finance over management by sustainability. All models have embedded value judgments [49], and the financial reporting model is no exception. Financial reports come with a built-in endorsement of certain behaviors. A simple example, adopted from Bayou et al. [50], illustrates the point. Accounting for profit is achieved through a simple identity familiar to anyone who has taken a beginners' accounting course, i.e., net income = revenues—expenses. There is a long history of arguments over what net income means and, consequentially, net income is defined by conventions that have become legalistic via the creation of standard-setting bodies. Revenues are those things defined by law and standards that meet the legal definition of revenues, notably, inflows of assets or reductions of liabilities that result from the provision of goods or services. Of course, "assets" and "liabilities" are formally defined by legal precedent and by standard-setting bodies. "Expenses" are likewise defined by law or standards, such that net income is a social construction that is designed to approximate what a company could distribute to owners in the form of a dividend and allow the company to sustain that dividend into the future. Conventional financial statements are thus based on a concept of sustainability, but sustainability confined to a narrow financial idea of sustainable consumption of the means of generating income (capital), i.e., the linear model of production of take, make, use, lose. Income comprises the amount of market value turned into the utilities of consumption without jeopardizing future periods of consumption. Implicit in these laws and standards are notions of economic causality such that the financial statements are useful for predicting what the differential consequences of actions might be.

As Bayou et al. show, we can relabel the simple income equation to illustrate what the financial performance of a corporation entails in terms of its various stakeholders. The net income of the corporation can be described as the "gross income of shareholders", since the net income of the corporation is defined by accounting rules as an addition to the equity of the shareholders. Accounting conventions thus portray the net assets of the corporation as being net assets of shareholders, implying an ownership relationship

between shareholders and those assets. However, what accountants label as stockholders' equity are not net assets of stockholders, but net assets owned by the corporation [30,31]. Accounting standard-setters have sown much confusion by eliding the ownership status of stockholders vis-à-vis corporations. Revenues are simply the resources the corporation is able to garner in the marketplace or take via exercising power (e.g., subsidies or preferred access to natural resources) through the provision of goods and services to customers. Expenses—regarded as negative elements vis-à-vis net income—are actually the incomes the corporation provides to its employees, its suppliers, its creditors, its government, and itself through capital allowances (depreciation). In addition, a corporation is a generator of "externalities", i.e., those costs and benefits inflicted on others for which there is only an accounting enacted through regulations designed to control those externalities (e.g., laws against emissions of harmful chemicals). Indeed, Greenfield [51] has described corporations as externality machines. The re-described income equation now takes the form:

Gross income of shareholders = revenues—gross incomes of employees, suppliers, creditors, and governments—capital charges—net externalities.

Gambling [23] entertained the possibility that all of these components of corporate income would lend themselves to economic valuation, and some of the early social accounting efforts were directed at estimating the monetary value of externalities. Were we able to do so, we would have an economic solution to making corporations internalize their negative externalities. The importance of internalizing externalities is that the dimensions of social performance epitomized by the GRI dimensions are categories of various externalities. Environment encompasses all of those things a corporation does that enhance or debase the physical environment. Labor encompasses not just what a company pays its employees but the safety of the workplace it provides, the level of stress it creates, the surety of employment it provides, etc. Society encompasses the corporation's activities as a member of the community—its citizenship. Human rights encompasses what the corporation does that respects basic human rights, such as respecting the rights of workers to organize. The conventional financial statements contain the dimensions of sustainability reporting but elide all those dimensions for which no ready economic value measure is available.

Up until now, sustainability reporting has consisted mostly of unstructured, nonmonetary indicators of performance with respect to what the company is doing to "internalize its externalities." For example, a company will report on its initiatives to reduce its carbon footprint and may even provide measures of how much $CO_2$ it has saved via these initiatives. It may provide a narrative about its efforts to contribute to the communities in which it has operations, such as the activities of its charitable foundation. What such reports lack is the structure that exists in the financial reports. Missing is the link of causality between the sustainability outcomes and the financial ones. Without such links, the value of reports as a tool for learning about a company's various effects is limited. Making such linkages more explicit is the purpose of the latest development in sustainability reporting: integrated reporting.

## 3. Integrated Reporting and Integrated Thinking: Proposing a Reporting Model

The newest development in the evolution of sustainability reporting is the concept of integrated reporting. A coalition of investors, regulators, NGOs, the accounting profession, and standard-setters has formed itself into the International Integrated Reporting Council, whose mission is to develop systems of reporting that communicate what companies are doing in terms of value creation [5]. The task the IIRC has set for itself is a system of reporting that dispenses with the prevalent system of disconnected pieces of reporting that have characterized sustainability reporting under such frameworks as GRI. A core idea of the proposed model of integrated reporting is the idea that corporations create value not just in terms of return to shareholders but in the form of various "capitals", i.e., financial, manufactured, intellectual, human, social and relationship, and natural [5] (p. 2). Integrated reporting is radical in the sense that it is predicated on the parallel concept of integrated thinking. Rather than the dimensions of social performance acting as constraints in an

optimization problem whose objective function is still financial, the dimensions of social performance (now designated as various kinds of "capital") operate as both constraint and objective. IIRC describes integrated thinking as follows (see Figure 1):

> *Integrated thinking is the active consideration by an organization of the relationships between its various operating and functional units and the capitals that the organization uses or affects. Integrated thinking leads to integrated decision-making and actions that consider the creation of value over the short, medium and long term* [5] *(p. 2).*

However, Feng, Cummings, and Tweedie [52] note how this definition of integrated thinking conveys little about what it means to various constituents. Notably, in August of 2019, 200 CEOs of firms comprising the Business Roundtable signed a declaration that shareholder value, as the singular goal of their companies, is detrimental to the long-run viability of their companies. As described in the New York Times report of the declaration: "No longer should the primary job of a corporation be to advance the interests of its shareholders" [53] (p. 1). CEOs of the world's largest corporations have publicly declared the need for integrative thinking. However, since the Business Roundtable was originally organized to counter the corporate social responsibility movement in the 1970s, there is skepticism in some quarters about whether the Business Roundtable declaration will result in any change [54,55].

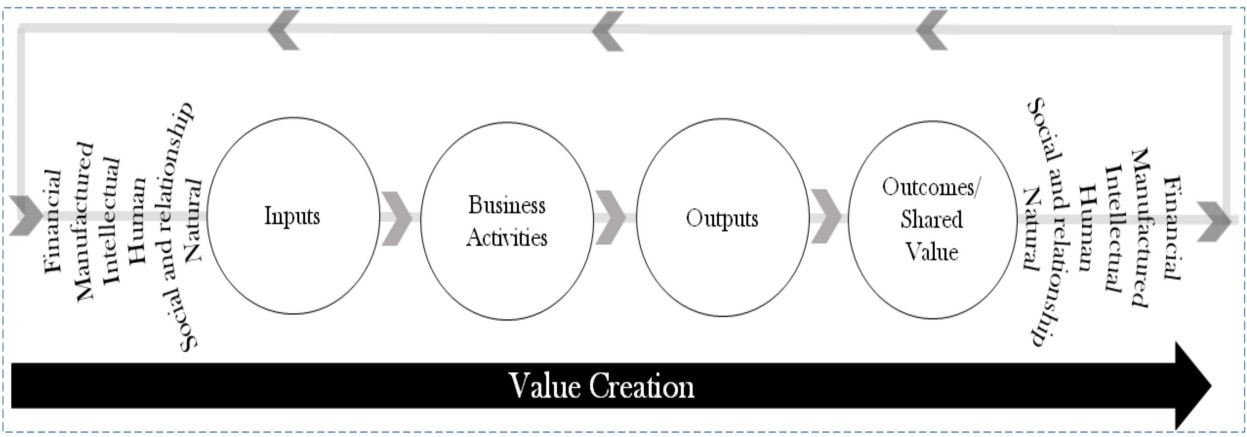

**Figure 1.** Value creation process.

The complexity of integrated thinking we illustrate via the simple identity we explicated in the previous section. As we have mentioned, early social reporting models entertained the idea that the economic valuation of all management actions was feasible. Clark Abt and Associates developed a set of financial statements based on its assessment of the economic value of its positive and negative externalities. Human resource accounting models developed during the 1970s were an attempt to capture the employee/employer relationship in financial terms to remind management of the value of its employees and the long-term value of investing in their skills and job satisfaction. However, the notion of integrated thinking and reporting on the results of such thinking turns a conceptually simple model into a remarkably complex one:

"Value" added for shareholders = "value" added for customers +/− (?) "value" added for employees +/− "value" added for suppliers +/− "value" added for creditors +/− "value" added for governments +/− "value" added for capital replacement +/− "value" added to environment, community, society, etc. This is illustrated in Figure 2.

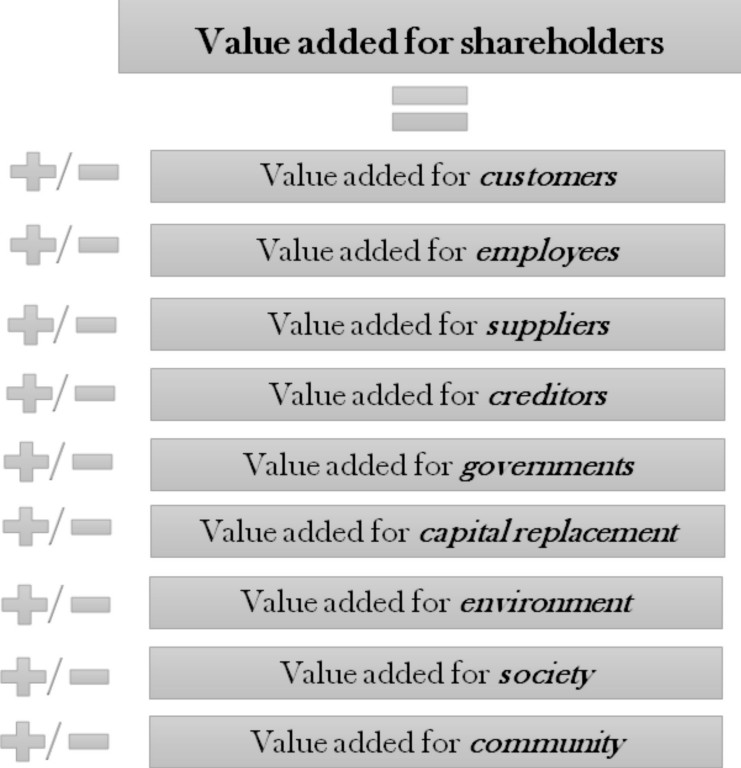

**Figure 2.** Value added for shareholders.

"Value" appears in ironic quotes since the idea of integrated reporting means that the role of management in thinking in an integrated way is to make decisions that involve the short-, mid-, and long-term development of values that are incommensurate. The structured system of financial reporting based on a system of double-entry bookkeeping is erected on representing the outcomes of business activity that meet more-or-less stringent criteria established in commercial law and customs. Assets may consist of all manner of incommensurate things (land, buildings, machines, patents, etc.) that are "measureable" in incommensurate ways (acres, pounds, units, etc.). Magically, all of these disparate things are aggregated into a meaningful total by the existence of certain social institutions— markets, money, property rights, enforceable contracts, exchange, etc. The idea of an entity or something with discernible boundaries is essential in order to establish just what the reports are "reports" about. The idea of accountability entertained by the system of reporting for business activity prevalent today is predicated on a rather limited notion of stewardship, mainly one of complying with society's laws and making a profit via the prudential use of resources provided by legal owners of the business. This traditional model of business management relies on the "rules" of the game to take care of the various values of capitals; management concerns itself only with the financial.

Though integrated reporting is described as a radical departure from the traditional model, it is less radical than it appears. Dumay, Bernardi, Guthrie, and La Torre [56] argue the IIRC is an example of regulatory capture: " . . . the evidence shows the substantial influence of large business and the profession" [56] (p. 465). As the IIRC describes the advantage of IR, it retains the conventional "decision usefulness" objective of financial reporting. It also focuses on capital providers as the principal beneficiaries of reporting: "Integrated Reporting (<IR>) promotes a more cohesive and efficient approach to corporate reporting and aims to improve the quality of information available to providers of financial capital to enable a more efficient and productive allocation of capital" [5] (p. 4). This is not a radical departure from Friedman's contention that firms should make as much money as they can; it varies only in stipulating that the time for doing so is not confined to the immediate moment. IR is another variation on the assumption that rationalized

corporate social responsibility in the 1970s that being good translated into being more profitable—making money by being good. Implicit in this argument was that if being good was not profitable, then being good was not necessarily an end in itself. This same rationale is evident as the IIRC explicates what represents "materiality" in what is reported:

> *"The ability of an organization to create value **for itself** (emphasis added) enables financial returns to the providers of financial capital. This is interrelated with the value the organization creates for stakeholders and society-at-large through a wide range of activities, interactions and relationships. When these are material to the organization's ability to create value **for itself** (emphasis added), they are included in the integrated report." [5] (p. 4).*

Though integrative reporting and integrative thinking may induce right actions on the part of management vis-à-vis employees, the environment, society, etc., the reason for these right actions is still based on economic calculation. The force that guides the behavior of corporations is still that of the capital providers and the validation of sustainability still comes from the belief that market forces should be relied on to guide human affairs better than any other "forces." However, it may not be sufficient for the legitimacy of business as an institution merely to do good acts, since the meaning ascribed to any action also involves a consideration of the reasons for the action. To be legitimate moral agents, corporations must have moral reasons for the actions they take [13].

Integrated thinking, on the other hand, has within it potential to transform business management into a genuinely stakeholder-centric practice. Integrated thinking is an idea that explicitly recognizes that the "rules of the game" are not exclusively to be relied upon to assure stewardship of the values of manufactured, intellectual, human, social and relationship, and natural capitals. Indeed the "rules of the game" is one of the capitals whose value needs enhancing. An integrative thinking management is one that accepts responsibilities that transcend the traditional role we ascribe to management [57–59]. The currencies with which these disparate capitals must be valued are different, do not have a rate of exchange, and will prove ultimately to depend on moral arguments for justification. By necessity, the firm becomes a moral agent. Dworkin [60] argues that "value" in any of its forms boils down to being just one thing: "Value" is not a scientific notion; in nature nothing is more "valuable" than anything else. Valuing is, perhaps, a uniquely human activity." Non-human beings exhibit "preferences", e.g., squirrels may choose sunflower seeds over hickory nuts when presented with a "choice." However, a preference does not mean a valuing process is performed. Preferences (tastes) are a kind of value, but they are not exclusively the basis of valuing. Humans may reasonably value something they do not prefer, but it is difficult to imagine squirrels doing that. The problematic for integrative thinking is that the various capitals, because they are incommensurate, require management to adopt a substantially different identity. None of the various capital values can take priority over any of the others. The traditional view of business practice is that when values "clash", financial value always trumps. That is, all other values are "valued" relative to financial value alone. This is the standard argument for social responsibility that by being "good" one can be more profitable. What happens when being "good" does not translate into being profitable? All too often, it means management chooses to do "bad". The Ford Pinto case is a vivid example. Ford managers knew the design flaw in the Pinto gas tank would result in 180 fiery deaths per year yet calculated that the cost of fixing the flaw was greater than the estimated cost of 180 accidents resulting in deaths [61]. According to Dworkin [60] (p. 101): " . . . interpretation knits values together. We are morally responsible to the degree that our various concrete interpretations achieve an overall integrity so that each supports the others in a network of value that we embrace authentically."

The implication of what is involved in knitting the disparate values of capitals is that any resulting "report" about the activities that produce them must constitute an integrated narrative about the firm. That is, the actions taken to enhance the value of social and relationship capital must constitute a coherent narrative when considered in

conjunction with actions taken with respect to the other capitals. Integrative reporting is reporting about a network of values not simply that of the financial value created during a certain time period by these various capitals. The value of integrative reporting as a goal for companies is that the act of trying to produce such a report will ultimately lead to management thinking and acting in ways that are conducive to a kind of stewardship more conducive to genuine sustainability. As our analogy in the introduction indicates, zinc and copper are two distinct metals analogous to financial reports and sustainability reports. However, these two separate metals can, by heating, be turned into something distinctly different from merely the sum of the two elements, i.e., brass. Integrative reporting means to represent a narrative about the firm that is not simply a financial representation and a sustainability one, but a substantively different story representing a new element of business conduct. Integrative thinking is the alchemy intended to make this happen.

## 4. Discussion

An important implication of integrative thinking is the prospect that it necessitates changes in the way business managers are educated. The prevailing "grand narrative" about business is one of competitiveness with a focus on maximizing one thing—shareholder value—via economically rational actions. This grand narrative is not adequate for the role of managers as integrative thinkers concerned with balancing numerous incommensurate values. There is obviously an issue of legitimacy as to whether business managers should have the power to decide and act to enhance the multiple values. It is a legitimate moral and political question as to whether business managers should be the members of society empowered to plot the human course on sustainability. It is quite likely that integrative thinking on the part of management will require alterations in important social institutions, e.g., changing corporate law to acknowledge that a corporation is not a private person and, therefore, has no human rights. We do not intend in this paper to suggest what institutional changes may be necessary to allow managers to think "integratively". Based on work by McCumber [62], Bayou et al. [50] discuss the issue of truth and ethics in accounting by proposing a perspective that could serve as a grand narrative more conducive to what the IIRC has in mind with respect to integrative thinking. Whether explicitly stated as such, corporate social responsibility/sustainability is essentially about the moral behavior of corporations. If the neoliberal model of self-interested action in markets leads to the best welfare outcomes (because they are most economically efficient) then the entire issue of CSR or sustainability would be moot. It is the very fact that the problems associated with the current economic order are so persistent that we conclude simple market logic is insufficient for dealing with these problems. The new kind of stewardship suggested by IIRC presumes the inadequacy of the narrative about business that justifies management action dedicated solely to the financial success of the corporate enterprise.

According to Bayou et al. [50], a way to think about a "true" accounting is to conceive of the role of accounting information as "situating" a firm along a temporal continuum. According to McCumber's notion of truth, truth is a temporal notion. Something is true as of a particular time. McCumber enunciates the important implication for integrative thinking:

"The world we live in does not mirror some given natural reality but is constructed out of it by various principles of selection and ordering; these principles fall under the overall heading not of faithful replication of the givens or truth, but of the regulated construction I call "situating"." [62] (p. 41).

Situating is the essential role of corporate reporting; that is, to tell the "truth" about a company at a particular moment in time is to develop a narrative that is both ordered and as comprehensive as possible. Ordering is essentially what the GRI dimensions or the IR's various capitals intend to accomplish. Comprehensiveness means to tell as much as you know. No one can know absolutely everything about the past or the present. Knowledge of the future is even more limited, but a comprehensive rendering requires that as much as we do know about where we are is what we tell.

Comprehensiveness may necessitate significant changes to the traditional financial reporting model. IR, as it is currently conceived, takes the financial reports and the value judgments embedded in them as a given and seeks to find ways to integrate information about the capitals into those financial reports. However, perhaps it is the financial reports that must be altered for better integration with the information about capitals. For example, in the United States, the FASB has a sister organization that issues standards of financial reporting for state and local governments—the GASB (Government Accounting Standards Board). The slow progress made to develop an integrated system of reporting is to an extent attributable to an inconsistency in how accountants think about different types of organizations. This is reflected in the narrative produced by the Government Accounting Standards Board (GASB), an organization in the U.S. created in the early 1980s to promulgate reporting standards for state and local governments. The GASB articulated in a policy white paper why governments and private firms should be accounted for differently. This was obviously necessary to justify the existence of GASB, since if there are no differences, the FASB would suffice. The irony of this position is that, subsequent to its creation, the GASB has promulgated standards for state and local governments that mostly parrot the standards for private corporations that were written by the FASB. Practically speaking, the GASB has imposed the same standards on state and local governments as the FASB has imposed on private corporations. The distinction GASB [63] (p. 1) made to argue that standards for governments would be different was:

> "*Separate accounting and financial reporting standards are essential because the needs of users of financial reports of governments and business enterprises differ. Due to their unique operating environment, governments have a responsibility to be accountable for the use of resources that differs significantly from that of business enterprises . . . Governmental accounting and financial reporting standards aim to address this need for public accountability information by helping stakeholders assess how governments acquired public resources and used them during the period or are expected to use them in the future*".

This is a distinction without a difference. All financial accounting systems have as their basic architecture the double-entry, accrual accounting system which is based on exercising control over the receipt and use of resources represented in monetary terms. As Ijiri [11] noted, this system is to reflect the status of relationships between an entity and those "outside" the entity. The logic of accountability is just as deeply engrained in the reporting rules for private enterprises as the GASB alleges is the case for governments. Writing an accounting rule inescapably creates something for which the entity subject to the rule is newly accountable [64]. The distinction the GASB draws that makes it believe private businesses are substantially different from governments is based on the outdated linear model of business we described earlier. A sustainable corporation is one that has the same responsibilities for using resources as does any government. Indeed, the power of corporations over the actual lived lives of people today exceeds that of most governments [65]. The moral basis of the idea of corporate social responsibility or corporate sustainability is the recognition of this power and how that power should be responsibly used [66].

Perhaps, then, the reporting model for governments is a tentative first step toward integrated reporting and should be adopted by corporations. The current lack of integration in reporting stems from the fact that the financial objectives of business are at odds with the sustainability elements. Profit comes first, which induces managers of corporations to externalize as many costs as they can. An economic system that is truly sustainable (Raworth's [58] doughnut economy) is one in which corporations are accountable for internalizing social costs and creating positive externalities. The externalities of the twenty-first century are more complex and perhaps less amenable to amelioration that those that concerned the early advocates of CSR. For example, Rauch [67] chronicles the effects of Facebook and the toxic effect it has had in perpetuating untruths, creating communities that pose a serious threat to democracy. To internalize social costs and create positive

externalities, managers and the stakeholders to which managers are accountable must have the information about resources deployed and outcomes achieved along all dimensions (whether it is the five capitals or some other "valuable" goals). The accountability model for government reporting is far more instructive than the traditional financial reporting model we use currently. The governmental reporting model, rather than focusing on net income, has the form: change in net assets = revenues and other sources—expenses. GASB requires governments to report expenditures by program or function, i.e., for what were resources used. These programs or functions are accounted for in different "funds." Applying this model to a sustainable corporation would entail reclassifying activities by dimensions of sustainable performance. A required report for government is to report the budgeted and actual amounts expended for each program. A very simple illustration of such a report for a sustainable corporation is shown in Table 1.

**Table 1.** Integrated report of expenditure-responsible corporation for the year ending 31 December 20xx.

| Capital | Budgeted | Actual | Variance |
|---|---|---|---|
| **Natural (Detailed)** | | | |
| *Zero hunger* | x | xx | x |
| *Clean water* | xx | xx | — |
| *Clean energy* | xxxx | xxx | (x) |
| *Sus. Cities* | NA | NA | — |
| *Climate action* | xxx | xx | (x) |
| *Life below $H_2O$* | NA | NA | — |
| *Life on land* | xx | xx | — |
| *Partnerships* | xx | xxx | x |
| **Financial** | | | |
| **Manufactured** | | | |
| **Intellectual** | | | |
| **Human** | | | |
| **Social and relationship** | | | |

For example, the IIRC provides classifications based on the six capitals that form the bases of performance that we listed earlier in the paper: natural, financial, manufactured, intellectual, human, and social and relationship. For each of these dimensions, there are also subcategories that we have related to the UN's seventeen sustainability goals (see Figure 3). Each of the goals would constitute a different unit of account or fund. A financial report based on this accountability scheme would provide financial measures of how much resources were used by each fund. Thus, just as governments disclose how much was spent for public safety, health and welfare, waste management, etc., the corporation would disclose what it spent in each fund to accomplish clean water, clean energy, etc. In Table 1, we provide the details for just the natural capital. These resource expenditures provide two kinds of information. One is a proportional representation of how resources are actually being used by the company, i.e., is the company putting its money where its mouth is? Secondly, the measures of performance along each of these dimensions may be assessed financially. What is the yield per dollar for each dimension? Each capital as proposed in the IIRC framework with its sub-goals constitutes a separate fund reporting the resources committed to that fund and the actual expenditure of resources to achieve that particular capital value. These resource expenditures would be further classified as operating or capital, such that capital assets committed to each "capital" would be distinguished. Thus, the financial information pertaining to cost is integrated with the nonfinancial information pertaining to the accomplishments associated with achieving the value of each capital. Claims that it is not possible to provide this information would indicate that a company's sustainability efforts are not genuine, since it would seem essential for sound integrative management to be able to equate financial resources consumed with results that are produced. Governments invest in public safety for the sake of public safety; there is no expectation that a positive rate

of return should accrue to the existence of a police force. Yet it is still sensible to evaluate what is accomplished as public safety in order to decide whether more resources should be devoted to that activity. If it is the case that sustainable businesses must develop various "capitals", it would seem to be the case that managers would need to know how much they have spent developing each capital and stakeholders would need to know how much the company has committed to developing each of the capitals. A reporting model like this for governments would be a starting place to reframe the management function toward focusing on developing equally different "capitals", all of which are competing ends and none merely means to one over-riding goal of profit maximization. Likewise, for stakeholders to be able to have any effect in holding companies accountable, they need to know how much in the way of resources has been committed. Though it is often assumed that accounting is about numbers, its real significance is in the words it attaches to those numbers. The traditional financial statement is a narrative about profit making. But the narrative for not-for-profits cannot be about profit making, but is about delivering benefits to citizens, in the case of governments, or to clients, in the case of eleemosynary organizations. Through such a system of reporting, the framing of the organizational narrative and the behavior of management changes to one in which the development of the capitals can become an end of the organization, rather than only a means to a strictly financial end. In the U.S., there is a growing interest in what is called a B Corporation. Some states now permit the organization of a business corporation whose explicit purpose is to focus on IR capitals as ends in themselves for the corporation.

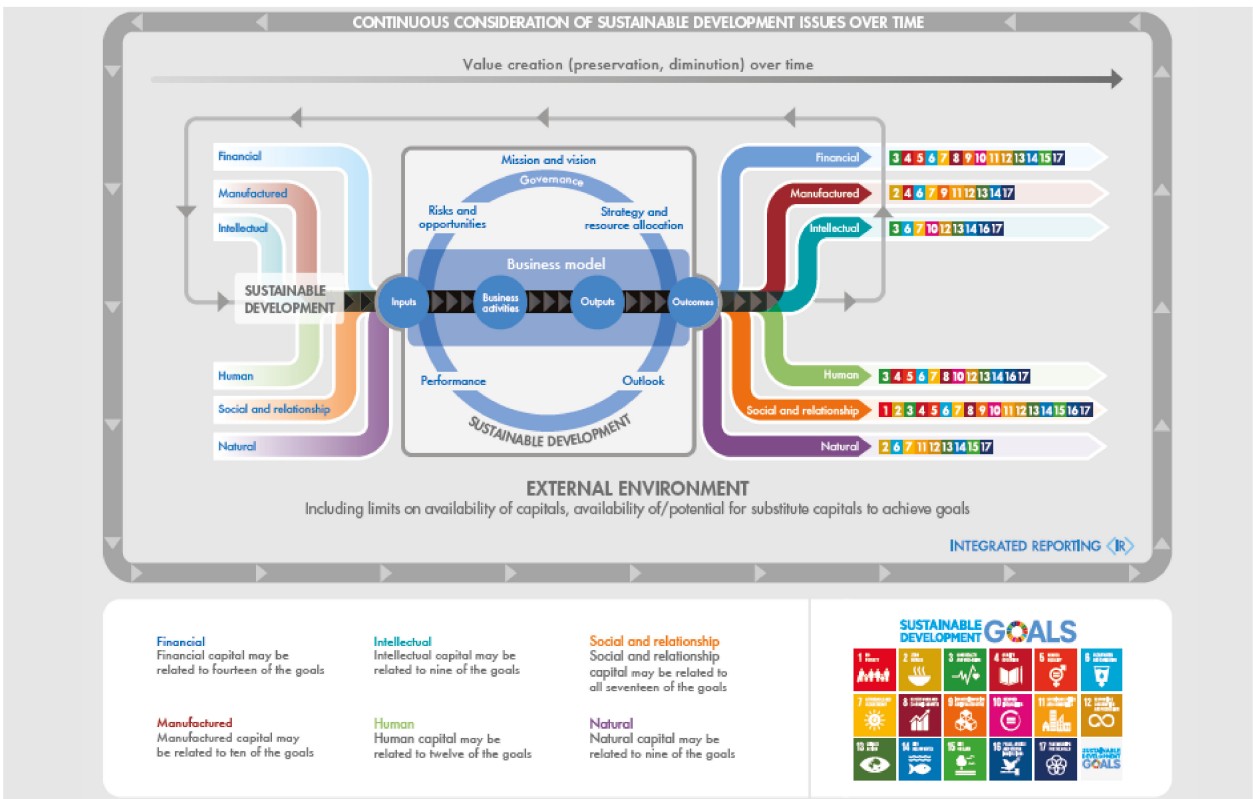

**Figure 3.** The link between the six capitals and sustainable developments goals. Source: Adams, C. [68]. The sustainable development goals, integrated thinking, and the integrated report: IIRC and ICAS.

## 5. Conclusions

Contemporary circumstances are radically different, e.g., the existential threat of climate change and the extreme concentration of wealth in fewer and fewer hands provides compulsion for the roles, impacts, and responsibilities of companies in society to change. The purpose of companies can no longer be limited to producing goods and services but

must also encompass creating sustainable value for themselves and society. The main motivation behind this is that companies' long-run prosperity is more dependent on working together with all their stakeholders for common purposes. Many large companies are coming to the awareness that focusing on merely short-run economic and financial results does not suffice for sustaining their existence. Hence, a growing number of corporations have started to share their nonfinancial information pertaining to their governance, their social effects, their environmental effects, and company strategies about how to address stakeholder concerns. The United Nation's Sustainable Development 17 Goals have also made it clear that the landscape of corporate reporting is changing. The 17 goals reflect a growing awareness that measures singularly focused on economic values are insufficient to explain the value a corporation adds to or takes from the welfare of society. It also reflects a growing awareness that the problem of corporate social responsibility may lie in the corporate form of business itself [51,66].

Given these developments, the sharing of financial and nonfinancial information in an integrated way to create a more comprehensive narrative of how companies create value has become an important factor in the business environment. This integrated approach [5] aims to be more comprehensively informative about firms' actions and thus induce firms to be more responsible to stakeholders other than just shareholders and creditors. The IIRC's integrated reporting model depends on the concept of integrated thinking and on telling a comprehensive value creation story of the firm. In addition to providing benefits to investors and other stakeholders, this approach provides a significant contribution to the company's self-development, in the form of conducting business based on integrated thinking.

The integrated reporting process advocated by IIRC is alleged to lead to strategic thinking aimed at sustaining the firm by creating multiple, incomparable "values." What remains unachieved is a genuinely integrated system of reporting on these values. The financial value achieved is reported elaborately in the traditional, highly structured financial statements, while the other values are reported together in a much less elaborate and unstructured sustainability report. An unstated premise of the IIRC's values (or capitals) is that corporations have a capacity for moral agency, since reporting about activities affecting employees, the environment, or the communities affected by the firm's actions opens the firm to potential moral blame from the general public. Stakeholders do not judge firms solely on how their actions affect financial performance, e.g., stakeholders would evaluate the dumping of toxins into a community's drinking water based not on how it improved the firm's financial performance, but on how it shows a disregard for other people.

For integrative reporting to lead to integrative thinking on management's part, the financial reports and sustainability reports cannot remain concerned with separate and unequal domains. We have proposed a tentative first step toward a truly integrated report based on the U.S. state and local governmental reporting model. It is common to analogize governments as being, after all, just businesses. However, if sustainable companies are to develop six different capitals, then, perhaps, the more appropriate analogy is that these companies are to act more like governments. Achieving the six values proposed by IIRC will require the commitment of resources. Just as state and local governments in the U.S. must report what resources they committed to achieving public safety, health and human welfare, education, infrastructure, etc., firms should report what resources they have committed to each of the six capitals. This is an explicit statement about each of the six capitals' relative value to the firm and a basis for assessing how the nonfinancial results achieved link to the resources committed to them. Such a model provides a more comprehensive "situating" of a firm vis-à-vis the allegedly equally important values.

Our analogy of putting an amount of copper and an amount of zinc into the same container yielding merely a container filled with copper and zinc is apropos of the current state of integrated reporting. However, applying the right amount of heat integrates these two elements and changes them into something different from merely their sum—they become a new substance, brass. If corporations are to be institutions contributing to the solutions to the major problems besetting society, then their story has to be constructed

not just of amounts of copper and amounts of zinc, but of brass—this seems to be what is required to tell a coherent story about corporate activity. A truly integrated reporting system conveys a story about how a company contributes to social well-being, not how it merely enhances shareholder value.

However new models of reporting might evolve to achieve more meaningful integration, the importance of situating to acting properly is that in order to act properly we need to know where we are. The capitals that comprise the various values integrative thinking intends to enhance imply a much more complex narrative about the "truth" of a company. Financial reporting is an extremely limited situation, indicating very little about anything other than general categories of assets, liabilities, equities, profits, cash flows, etc. Even at that, these reports are often woefully poor, since such reports have frequently failed to indicate immanent financial failure. Enron is the classic example. It exploited mark-to-market accounting to inflate its profits and utilized gaps in consolidation rules to keep billions of contingent liabilities off of its books [69]. Integrative reporting means situating the company with respect to a significantly larger domain of responsibility than mere financial performance. Thus, to think integratively in order to report integratively, management will have to develop a narrative about the company that is far more comprehensive than previously.

Situating a sustainable company to develop multiple, incommensurable capital values would seem to necessitate much greater focus on relationships. McCumber [60] describes the human world as analogous to nested Russian dolls. Individuals are nested within communities, which are nested within institutions, which are nested within societies, which are nested within a biosphere. Telling the "truth" about a company entails a comprehensive narrative about how a company adds "value" to these various "capitals". So, for example, what may be the truth about taxes in the current grand narrative about the corporate role is that through the cunning exploitation of incorporation law, transfer pricing practices, and international differences in taxation, the company paid little to no taxes. However, the consequences of not paying taxes affects the value of the other capitals in ways that could be significant. For example, education is perhaps the single most important thing for enhancing the quality of individual humans' lives. Its importance to society lies in the fact that in most places in the world education is a public good. Taxes are the primary financier of public goods, so to the extent that taxes are not paid, human capital development via education suffers. Managers thinking integratively consider the consequences of taxes along all dimensions of value before they decide what sustainable actions to take. A budget is a moral document not just for governments, but for any organization, since it reflects what the organization values. This is the genuine promise of integrative reporting, because it changes in a fundamental way how we consider the role of management in a world that may have already changed such that our previous models of how the world works are no longer relevant.

**Author Contributions:** Conceptualization, G.A. and P.F.W.; Methodology, G.A. and P.F.W.; Writing—original draft, G.A. and P.F.W.; Writing—review & editing, G.A. and P.F.W. All authors have read and agreed to the published version of the manuscript.

**Funding:** This research received no external funding.

**Institutional Review Board Statement:** Not applicable.

**Informed Consent Statement:** Not applicable.

**Data Availability Statement:** Not applicable.

**Conflicts of Interest:** The authors declare no conflict of interest.

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
