# Peer review of "Integrated Reporting and Integrated Thinking: Proposing a Reporting Model That Induces More Responsible Use of Corporate Power"

_sustainability, doi:10.3390/su14063277_

Round 1
Reviewer 1 Report
- What is the main question addressed by the research?
Working in dortis Integrated reporting and integrated thinking: proposing a reporting model that encourages more responsible use of corporate power. There is no clearly formulated purpose of the analyzes, both in the abstract and in the introduction. It is necessary to include the sentence: "The purpose of the presented analyzes is ..." Due to the lack of a clearly defined goal, it is difficult to clearly state whether the goal of the work has been achieved.
- Do you consider the topic original or relevant in the field? Does it
address a specific gap in the field?
The article is part of the current trend of corporate social responsibility and is a continuation of the discussion on the issues of corporate social responsibility.
- What does it add to the subject area compared with other published
material?
The material is a good systematization of the problems of corporate social responsibility and in this respect is a valuable material.
- What specific improvements should the authors consider regarding the
methodology? What further controls should be considered?
The methodology was correctly selected to conduct research in this type of problem. I have no objections.
- Are the conclusions consistent with the evidence and arguments
presented and do they address the main question posed?
The conclusions are correctly formulated and result from the conducted analyzes. As already mentioned in the first paragraph, the purpose of the analyzes needs to be clearly formulated, which will allow to fully define the achievement of this goal.
- Are the references appropriate?
The literature cited in the work corresponds to the current research status in the field of the discussed issues.
- Please include any assitional comments on the tables and figures.
No critical remarks regarding tables and figures.

Author Response
Mid-August of 2019 200 CEOs of firms comprising the Business Roundtable signed a declaration that shareholder value, as the singular goal of their companies, is detrimental to the long-run viability of their companies. As described in the New York Times report of the declaration: “No longer should the primary job of a corporation be to advance the interests of its shareholders” (Yaffe-Bellany & Gelles, August 19, 2019, p. 1). CEOs of the world’s largest corporations have publicly declared the need for integrative thinking.

Reviewer 2 Report
I would like to thank the editor of Sustainability for providing me a possibility to review this article manuscript on Integrated CSR reporting. I consider this article in general fitting the scope of the journal. In addition, I consider that the manuscript is close being publishable. The comments I have listed below can be considered minor and mainly depict the need to increase the level of critical thinking in this paper. I understand that this is the first attempt to conceptualize their model. Still, the paper is somewhat non-critical and too straight-forward in some parts of the text.
1) The authors mention the statement by the Business Roundtable. Maybe the authors could discuss that statement from a bit more critical perspective. Yes, the CEO’s have publicly shared their visions, but how credible is this statement eventually? https://hbr.org/2019/08/is-the-business-roundtable-statement-just-empty-rhetoric & https://www.forbes.com/sites/bobeccles/2020/08/19/an-open-letter-to-the-business-roundtable-181/?sh=71db90404001. Is this just an example of “CSR reporting” that is possible because of non-standardized forms of sustainability reporting. In other words, it is to list such statements, when the processes of reporting are not uniform.
2) authors mention Ford Pinto case as an example (p. 12). Maybe not all authors are familiar with the case and a reference could be added? Similarly at page 17 Enron is mentioned as a classic example.. on what? I might have a hunch on that but maybe not all readers are familiar with it? Hence a reference is needed.
3) Raworth’s doughnut economy – maybe a reference here also (p. 14)?
4) Figure 3 should be a Table (p. 14)
5) Figure 4: maybe the authors should elaborate more how they related the 17 UNSD Goals to their model? I think that this is the most important thing to focus on when developing further this manuscript. At the moment I do not fully understand their claims on the links. For instance SDG8 (on decent work and economic growth) how does that not relate to “Intellectual” outcome? For instance ILO discussed all UNSDG’s from the perspective of decent work: https://www.ilo.org/global/topics/sdg-2030/lang--en/index.htm Maybe the authors could consider this aspect to be added to the discussion?
6) On nested Russian dolls (p. 18): is that a good example? The nested doll represents an individual in which there is an individual (and so on…) Now, given the holistic approach in this manuscript this example might lead to wrong directions. Should we consider the doll from an individual perspective, i.e. one person is nested with a community and a society and so on.. do we all as individuals form one doll and so on? How many individuals fit into one doll in this fragmented world?
7) there are some examples in the text where the full reference is missing (like when making a reference on McCumber the year is missing (p. 18)).
Author Response

(The authors gave the same response as above.)
